# Evaluation of the Efficacy of Immersive Virtual Reality Therapy as a Method Supporting Pulmonary Rehabilitation: A Randomized Controlled Trial

**DOI:** 10.3390/jcm10020352

**Published:** 2021-01-18

**Authors:** Sebastian Rutkowski, Jan Szczegielniak, Joanna Szczepańska-Gieracha

**Affiliations:** 1Department of Physical Education and Physiotherapy, Opole University of Technology, 45-758 Opole, Poland; j.szczegielniak@po.opole.pl; 2Department of Physiotherapy, University School of Physical Education in Wrocław, 51-612 Wroclaw, Poland; joanna.szczepanska@awf.wroc.pl

**Keywords:** virtual reality, VR, COPD, pulmonary rehabilitation, anxiety, depression, stress

## Abstract

Anxiety has been estimated to occur in 21–96% and depression in 27–79% of patients with chronic obstructive pulmonary disorder (COPD). We found a scarcity of literature providing evidence on how virtual reality (VR) therapy affects the intensity of depressive and anxiety symptoms and stress levels in COPD patients undergoing in-hospital pulmonary rehabilitation (PR). This study enrolled 50 COPD patients with symptoms of stress, depression, and anxiety, randomly assigned to one of two groups. The two groups participated in the traditional PR programme additionally: the VR-group performed 10 sessions of immersive VR-therapy and the control group performed 10 sessions of Schultz autogenic training. Comparison of the changes in stress levels and depressive and anxiety symptoms was the primary outcome. Analysis of the results showed a reduction in stress levels only in the VR-group (*p* < 0.0069), with a medium effect size (d = 0.353). The symptoms of depression (*p* < 0.001, d = 0.836) and anxiety (*p* < 0.0009, d = 0.631) were statistically significantly reduced only in the VR-group, with a strong effect size. The enrichment of pulmonary rehabilitation with immersive VR therapy brings benefits in terms of mood improvement and reduction in anxiety and stress in patients with COPD.

## 1. Introduction

Chronic obstructive pulmonary disease (COPD) is a major threat to public health and is recognized as one of the most impactful of the common chronic diseases, making it the second most common cause of disability and the third leading cause of death globally [1]. The Global Initiative for Chronic Obstructive Lung Disease (GOLD) defines COPD as a disease state characterized by airflow limitation, causing shortness of breath, significant systemic effects involving the lung and extrapulmonary adverse reactions, with a high disease rate, high disability rate, high mortality rate and a long course of disease [2]. COPD is not only characterized by shortness of breath, chronic coughing, and sputum production, but also by a reduction in physical capacity and level of physical activity. The main risk factor for COPD is smoking, which in turn is correlated with the occurrence of depression; both frequently potentiate each other [3,4]. Prevalence studies indicated that patients with COPD are four times more likely to develop depression than those without COPD. Unfortunately, such patients rarely receive the appropriate comprehensive treatment [5]. Anxiety and depression have been found to increase the number of acute exacerbations and hospitalizations (thus reducing quality of life), weight, appetite, sleep disorders, fatigue or loss of energy, lack of concentration, pessimism about the future and suicidality [6,7,8]. Anxiety has been estimated to occur in 21–96% and depression in 27–79% of patients with COPD, values that are higher than for the general population or other chronic diseases [9].

In view of the serious morbidity and mortality of COPD, and its economic burden, pulmonary rehabilitation has become an essential component in its management. The adopted models of pulmonary rehabilitation vary in terms of intensity, duration, therapeutic methods and the form of physical activity taken by the patients. In recent studies, the impact of several weeks of hospital rehabilitation on the improvement of lung ventilation, exercise capacity, physical fitness and quality of life of patients with COPD has been shown [10]. To date, there is very little understanding about the potential of pulmonary rehabilitation to resolve the presence of anxiety and/or depressive disorders; however, previous evidence suggests that pulmonary rehabilitation may confer beneficial effects on anxiety and depressive symptoms in this patient group [9,11]. The majority of randomized trials consistently favoured interventions that combined exercise training with a psychological intervention for dyspnoea, anxiety, depression, quality of life, and exercise performance when compared with control or active comparator conditions [12]. Thus, there is a need to determine efficient methods for managing depression and anxiety in patients with COPD.

In 2018, a literature review evaluated the effectiveness of virtual reality (VR)-based interventions for symptoms of anxiety and depression. A meta-analysis based on 39 randomized controlled trials compared a VR intervention group (*n* = 869) and a control or active treatment group (*n* = 1122). Among the evaluated mental conditions, 31 studies concerned anxiety and anxiety-related disorders. The most frequently used VR device was the head-mounted display (35 studies). The authors concluded that patients exposed to a VR-enhanced intervention showed better improvement than those treated traditionally and thus VR therapy could be an effective choice for both clinicians and patients [13]. However, no articles in the review focused on patients with COPD. Thus, we found a scarcity of literature evaluating how VR therapy affects the intensity of depressive and anxiety symptoms and stress levels in COPD patients undergoing pulmonary rehabilitation in hospital. We therefore aimed to determine whether the implementation of immersive virtual reality during a pulmonary rehabilitation programme would produce a more efficiency in reduction in symptoms of depression and anxiety, as well as stress level in patients with COPD.

## 2. Methods

### 2.1. Participants

This study enrolled 50 patients diagnosed as having COPD from the Specialist Hospital in Głuchołazy, Poland, between October 2020 and November 2020. All patients gave written informed consent prior to their participation in the study. Patients who meet the inclusion criteria were randomly assigned to one of two groups (VR group or control group). Randomization was performed using the Research Randomizer (ratio 1:1), a web-based service that offers instant random assignment. Sealed envelopes were used for group assignment. Table 1 presents the characteristics of both groups. The inclusion criteria were as follows: diagnosis of COPD; age 45–85 years; pulmonary rehabilitation conducted in a ward setting; and an anxiety or depressive symptom score of >8 on the Hospital Anxiety and Depression Scale (HADS). The exclusion criteria were as follows: cognitive impairment (Mini-Mental State Examination score of <24); inability to self-complete the research questionnaires; presence of disturbances of consciousness, psychotic symptoms or other serious psychiatric disorders at the time of examination or in the medical data; initiation of psychiatric treatment during the research project; contraindications for VR therapy (epilepsy, vertigo, eyesight impairment); and the patient’s refusal at any stage of the research project. The study adhered to the Declaration of Helsinki guidance, ethical approval was obtained from the Research Ethics Committee of the University School of Physical Education in Wrocław, Poland (Resolution No. 18/2020) and the study was registered in ClinicalTrials.gov (NCT04601545). The data collected from human subjects were collected, managed, and protected by the research team and hospital staff members. This trial was designed as an assessor-blinded, parallel group study.

### 2.2. Measures

Comparison of the changes in stress levels and depressive and anxiety symptoms was the primary outcome. As a secondary outcome, we evaluated functional capacity.

#### 2.2.1. Perception of Stress Questionnaire

The Perception of Stress Questionnaire (PSQ) is a 27-item scale with scores of 1–5 for each item: 21 items examine the level of stress in the area of emotional tension, external stress and intrapsychic stress; and 6 items refer to the lie scale. The total score for perception of stress ranges from 21 to 105, with a cut-off point of 60 for a high level of perceived stress. The questionnaire contains both positively and negatively formulated items in order to reduce acquiescent bias. Each item is answered using a four-point Likert-type scale ranging from 1 (‘almost never’) to 4 (‘almost always’): the higher the score, the greater the sense of stress [14].

#### 2.2.2. Hospital Anxiety and Depression Scale

The HADS is a 14-item scale that scores 0–3 for each item. The first seven items relate to anxiety (HADS-A) and the remaining seven items relate to depression (HADS-D). The total score ranges from 0 to 42, with a cut-off point of 8/21 for anxiety and 8/21 for depression: the higher the score, the greater the anxiety or depressive symptoms. The HADS was considered to be a valid research method [15].

#### 2.2.3. Functional Capacity

The evaluation of functional capacity included exercise capacity (6 min walk test: 6MWT) and lung function (forced expiratory volume for 1 s: FEV1). The 6MWT is a valid and reliable measure of exercise capacity for people with chronic lung disease [16]. The test is performed on a firm, flat surface over a 30 m section and measures the distance a patient is able to walk over a total of 6 min. The patient was told to walk as far as possible, being allowed to self-pace and rest as needed whilst traversing back and forth along a marked walkway. The test was performed according to European Respiratory Society/American Thoracic Society guidelines. The FEV1, which is important for evaluating COPD and monitoring progression of the condition, was expressed as a percentage of the forced vital capacity (FEV1) [17]. We calculated predicted values based on age, weight and height [18].

### 2.3. Instruments

A VR TierOne device (Stolgraf^®^, Stanowice, Poland) was used as the VR source. A head-mounted display and the phenomenon of total immersion created an intense visual, auditory, and kinaesthetic stimulation. The primary aim of the software was to calm the patient down and improve his or her mood. Additional aims of the software were to help patients regain emotional balance, allow them to recognize their resources for gaining power in the rehabilitation process, and trigger natural recovery mechanisms. The software features a Virtual Therapeutic Garden and is based on the Ericksonian psychotherapy approach. The garden is a metaphor for the patient’s health: at the beginning it appears as untidy and grey (Figure 1), yet with each session it becomes more colourful and alive (Figure 2), thus symbolizing the process of recovery of energy and vigour [19].

The content of the VR therapy was developed by Joanna Szczepańska-Gieracha, a certified European Association of Psychotherapy therapist, who specializes in the treatment of people with health problems. The project was also supervised by Krzysztof Klajs, the Director of the Polish Milton H. Erickson Institute, who is a professional supervisor and chairman of the Scientific Section of Psychotherapy of the Polish Psychiatric Association. The software was developed with a grant from the Polish National Center of Research and Development (POIR.01-02.00-00-0134/16).

### 2.4. Procedure

The two groups participated in the traditional pulmonary rehabilitation programme. Components were performed once a day, each for 15–30 min (depending on the task), five times a week for two weeks. Exercises were performed as follows: fitness exercises while standing, on the knees and lying on the side, abdomen and back; strengthening exercises of the diaphragm with resistance; prolonged exhalation exercise; chest percussion; inhalation with a 3% NaCl isotonic solution administered via an ultrasonic device; and stationary cycle ergometer exercise to obtain a training heart rate according to GOLD spirometric stages. The pulmonary rehabilitation programme has been described in detail in previous studies [20]. The difference between the groups is in the type of relaxation training: the VR group performed 10 VR therapy sessions of 20 min and the control group performed 10 Schultz autogenic training sessions of 20 min. See Figure 3 for the CONSORT flow diagram.

### 2.5. Statistical Analysis

All statistical analyses were performed using Statistica 13 software (StatSoft, Cracow, Poland). The statistical significance level was set at α = 0.05. Variable analysis was performed and, after testing, the normality of sample size distribution was verified with the Shapiro-Wilk test. Categorical variables were presented as numeric values and percentages. Continuous variables were presented as mean ± standard deviation (SD). To compare differences between the examined groups, we used Spearman’s rank test for categorical variables and the Mann-Whitney U test for continuous variables. Differences between variables concerning levels of stress, anxiety, depression, and functional capacity were compared using repeated-measures analysis of variance (ANOVA). The size of the between-group effects was determined by Morris effect size *d* [21] and classified as follows: 0.1–0.3, small effect; 0.3–0.5, intermediate effect; and ≥0.5, strong effect [22]. A stepwise linear regression analysis was performed to identify independent determinants of depressive and anxiety symptoms and stress levels. Two models were built: the first model considered change in depressive and anxiety symptoms as the dependent variable; the second model included change in stress levels as the dependent variable. For both models, potential determinants included the results of functional capacity, pulmonary function, group (VR or Control Group), and time point (pre-post rehabilitation). The sample size was calculated based on previous studies according to the effectiveness of immersive VR used to facilitate the relaxation process in stressed and anxious subjects with an effect size of 0.234; it was determined that 50 patients should be enrolled [23]. G*Power 3.1.9 software was used to calculate the sample size. Calculation was based on repeated-measures ANOVA: the within–between interaction type I error rate was set at 5% (α = 0.05), the effect size of the main outcomes was 0.234, and the type II error rate gave 90% power for the two groups and two repeated sets of measurements; correlation among the repeated measures was 0.5 and the non-sphericity correction ε was 1.0.

## 3. Results

None of the sociodemographic characteristics differed among the groups before the intervention (Table 1). The analysed data were obtained from 50 patients.

### 3.1. Evaluation of Stress Levels

Analysis of the PSQ within all domains did not reveal statistically significant differences between the tested groups at baseline. Within-group analysis of the VR and control groups showed a statistically significant improvement in emotional tension (*p* < 0.0003), external stress (*p* < 0.0092), and total score (*p* < 0.0069) in the experimental group when the rehabilitation programme was completed (Figure 4) (Table 2).

### 3.2. Evaluation of Depression and Anxiety

Analysis of the HADS showed a statistically significant difference in the general HADS scores at baseline between the groups (*p* < 0.0198), indicating that patients in the VR group suffered more symptoms of depression and anxiety. Within-group analysis showed a statistically significant improvement in the HADS-A (*p* < 0.0009), HADS-D (*p* < 0.0001) and general HADS (*p* < 0.0001) scores in the experimental group after completion of the rehabilitation programme, but there were no statistically significant changes in the control group (Figure 5) (Table 2).

### 3.3. Evaluation of Functional Capacity

Analysis of the exercise capacity and pulmonary function baseline characteristics between groups showed statistically significantly lower FEV1 values in the experimental group (*p* < 0.0230). Within-group analysis showed a statistically significant improvement in exercise capacity for the experimental group (*p* < 0.0018) and control group (*p* < 0.0002), as well as an improvement of FEV1 in the control group (*p* < 0.0429), after completion of the rehabilitation programme (Figure 6) (Table 2).

Two multivariable regression analyses were performed to identify determinants of the depressive and anxiety symptoms and stress levels. In the first model, which considered stress levels, 6MWT and FEV1 were independently associated with an improvement in the PSQ total score (Table 3). In the second model, considering depressive and anxiety symptoms, time point (Pre-Post rehabilitation) was independently associated with an improvement of HADS total score (Table 4).

## 4. Discussion

Concerning our primary outcome, the results for the VR group show that there were large reductions from pre- to post-rehabilitation in the total score stress level (*p* < 0.0069), including emotional tension (*p* < 0.0003) and external stress (*p* < 0.0092). Data analysis of these domains revealed an intermediate effect size, whereas for intrapsychic stress, the effect size was small. Moreover, the VR group achieved a statistically significant reduction in symptoms of anxiety (HADS-A: *p* < 0.0001), depression (HADS-D: *p* < 0.0009), and general HADS scores (*p* < 0.0001). What should be emphasized is that the VR group achieved large effects in the therapy of depression and anxiety compared with traditional Schultz autogenic training. The differences prove that the levels of anxiety and depression in the group undergoing VR training diminished, whereas in the control group, the levels tended to increase (Figure 3). In our opinion, these are the key findings highlighting the need to introduce VR therapy in patients with symptoms of anxiety and depression. Moreover, we found an association between changes in stress level and functional capacity. Patients with poorer 6MWT and FEV1 values presented higher stress levels. Regarding the predictors of depressive and anxiety symptoms, no association between HADS score and functional capacity was noted. Only a very poor fit of the regression model to the time point of measurements was noted, which indicates that after the rehabilitation was completed, the patients presented reduced depression and anxiety symptoms.

In recent studies, clinical anxiety has been recognized as a significant problem in COPD, with an estimated prevalence of up to 40% [7]. Patients with symptoms of anxiety and depression lack faith in the success of rehabilitation and present unwillingness to work on improving health. This alone causes a decrease in the effectiveness of rehabilitation—patients with depression achieve 50% worse treatment results, which causes further deepening of depression and pushes patients into a closed circle of infirmity and lack of motivation. To date, only one study has been conducted on the effectiveness of a VR-based pulmonary programme in patients with COPD on anxiety. Jung et al. provided a VR headset with a VR app and a Nonin 3150 oximeter probe at home for an 8-week period [24]. Pulmonary rehabilitation in VR was designed to enable patients to perform exercises in their homes for at least 20 min per day. The rehabilitation comprises physical exercises led by a virtual instructor in the form of a 3D avatar. The physical exercises were drawn from the traditional pulmonary rehabilitation programme and tailored to be suitable for patients while wearing the VR headset (e.g., seated exercises). Quantitative analysis revealed an intermediate effect size (d = 0.331) for anxiety symptom reduction. However, it seems that the achieved effect was related to the fact that the patient performed regular exercise (regular physical activity) rather than being due to the therapeutic effect of the application itself.

Studies have shown that anxiety and depression in COPD patients during non-VR pulmonary rehabilitation can be reduced. According to Gordon et al.’s meta-analysis, the pooled standard mean difference (SMD) effect appears moderate for the outcome of anxiety symptoms after pulmonary rehabilitation (SMD = −0.53; 95%CI = −0.82 to −0.23), equating to a mean difference of −2.2 for HADS-A, and large for the outcome of depressive symptoms (SMD = −0.70), with a mean difference of −2.5 for HADS-D [9]. The results of Coventry et al.’s meta-analysis indicated significant improvements in depression (SMD = −0.47; 95%CI = −0.66 to 0.28) and anxiety (SMD = −0.45; 95%CI = −0.71 to 0.18) only in a multicomponent pulmonary rehabilitation intervention. Relaxation, emotional support, and cognitive therapy were associated with small but non-significant reductions in depression and anxiety. Remarkably, when the analysis was restricted to the trials that included both psychological and exercise components, the effect size increased to 0.64 for depression and to 0.59 for anxiety [25]. These results are in line with our study. Analysis of our results showed even greater effect sizes of 0.84 for depression and 0.63 for anxiety. In our opinion, the results were determined by applying VR as a therapeutic method. The effectiveness of immersive VR therapy in reducing anxiety and depression has been shown within other fields of rehabilitation. Moreno et al. noticed a reduction in psychological aspects after VR interventions targeting cognition within patients with neurocognitive disorders [26]. Won et al. stated that VR could help in the treatment of paediatric chronic pain via neuromodulation, as well as physical therapy [27]. Tennant et al. concluded that the use of immersive VR in clinical oncology settings improved patient well-being in terms of mood, anxiety, and pain [28].

Our findings are in line with the recent meta-analysis by Li et al., who evaluated the effect of mind-body exercise on COPD patients with anxiety and depression: 13 studies showed mind-body exercise to have significant effects on reducing depression (SMD = −0.86, 95%CI = −1.14 to −0.58; *p* = 0.000, I2 = 71.4%), and 11 studies demonstrated significant effects on reducing anxiety (SMD = −0.76, 95%CI = −0.91 to −0.60; *p* = 0.04, I2 = 47.4%) [6].

Regarding our secondary outcomes, the results indicate an improvement in exercise capacity in both groups undergoing a two-week intensive pulmonary rehabilitation programme. Clinically insignificant changes in the FEV1 in both groups were also shown, which is in line with the widely described results, suggesting minor changes in obstruction after short-term pulmonary rehabilitation. Nevertheless, the most important aspect is the way in which a patient functions after rehabilitation is completed. Under hospital conditions, patients are motivated daily by medical staff to participate in pulmonary rehabilitation and, as the results suggest, all patients achieve significant improvements in cardiopulmonary function. The complications commence upon returning home. If the symptoms of anxiety and depression are not reduced, the patient quickly returns to her/his former sedentary lifestyle and deteriorating health condition by lacking the motivation and energy for further individual rehabilitation in the form of regular physical activity. Therefore, in our opinion, improving mood and reducing the level of anxiety and stress is a key task of pulmonary rehabilitation in order to increase the chance of long-term health improvement and maintain the beneficial effects of pulmonary rehabilitation after the patient leaves the hospital ward.

The results of this study may be relevant for the pulmonary rehabilitation of patients with COVID-19. An international task force (ITS), including the European Respiratory Society (ERS) and key opinion leaders from the American Thoracic Society (ATS), as well as key opinion leaders in the field of lung rehabilitation was established. ITS stated that common symptoms reported one year later by intensive care unit survivors, including patient with acute respiratory distress syndrome (ARDS), include anxiety (34%) and depression (33%). The majority of the experts recommended strongly (71%) or conditionally (24%) for COVID-19 survivors with symptoms of psychological distress at 6–8 weeks after discharge from the hospital receiving a formal psychological assessment [29,30,31,32].

To the best of our knowledge, our study is the first to explore the effectiveness of immersive VR Therapy on anxiety, depression, and stress among patients with COPD. Although this study provides clear evidence, we recognize that some limitations should be considered. Firstly, the PSQ was used as a secondary outcome measure but, despite its high repeatability and validity, it is not widely used in scientific research. Secondly, future studies could be enhanced with a wider range of diagnostic tools, including more objective ways of measuring stress levels (e.g., cortisol levels). Thirdly, the lack of inclusion of full spirometry examination data in the analysis may cause bias in the results. Follow-up assessment could provide additional valuable information on comparing effectiveness with traditional therapeutic methods.

## 5. Conclusions

The coexistence of anxiety and depression symptoms is a common and serious problem in patients with COPD that aggravates prognosis and increases mortality in this group of patients. The enrichment of pulmonary rehabilitation with immersive VR therapy brings benefits in terms of mood improvement and reduction in anxiety and stress in patients with COPD. VR therapy is more effective than the traditionally used Schultz autogenic training. Further follow-up studies with long-term observation of patients after the end of hospital pulmonary rehabilitation are necessary to assess the sustainability of the effects obtained during the stay in the rehabilitation ward.

## Figures and Tables

**Figure 1 jcm-10-00352-f001:**
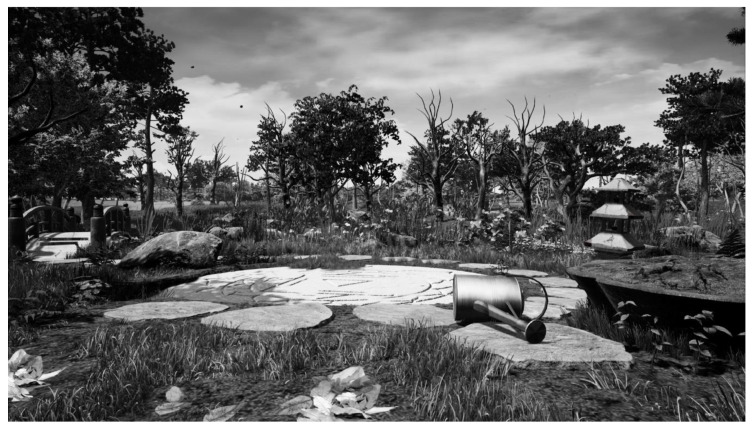
Virtual garden at the initial stage of therapy.

**Figure 2 jcm-10-00352-f002:**
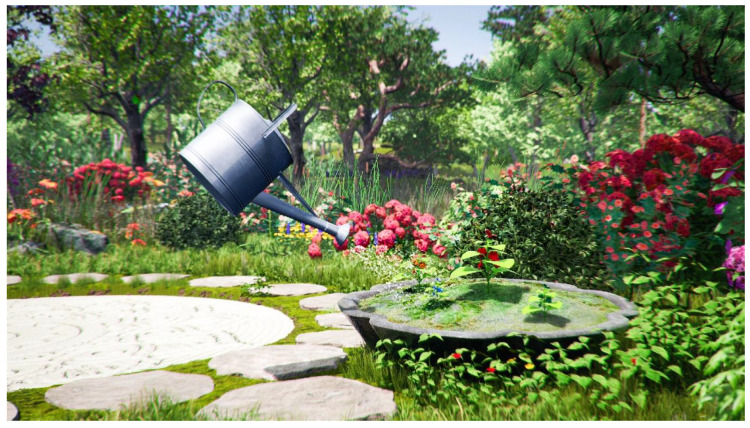
Virtual garden at the final stage of therapy.

**Figure 3 jcm-10-00352-f003:**
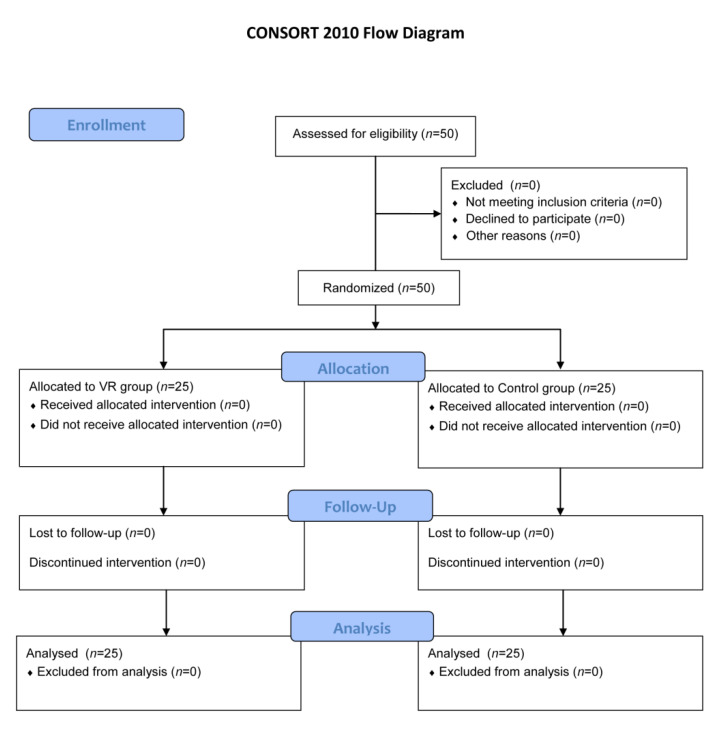
CONSORT flow diagram.

**Figure 4 jcm-10-00352-f004:**
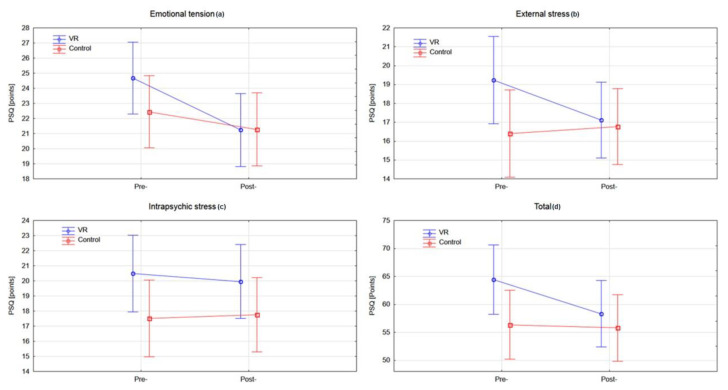
Analysis of the PSQ questionnaire in both groups pre- and post-rehabilitation program: (**a**) emotional tension (**b**) external stress (**c**) intrapsychic stress (**d**) total score. VR: virtual reality; PSQ: Perception of Stress Questionnaire.

**Figure 5 jcm-10-00352-f005:**
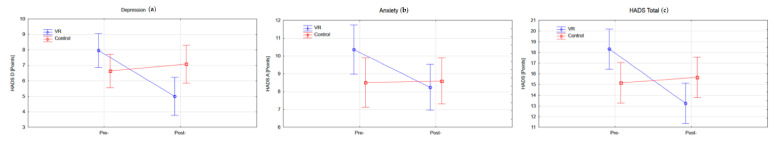
Analysis of the HADS (Hospital Anxiety and Depression Scale) questionnaire in both groups pre- and post- rehabilitation program: (**a**) depression, (**b**) anxiety, and (**c**) total score.

**Figure 6 jcm-10-00352-f006:**
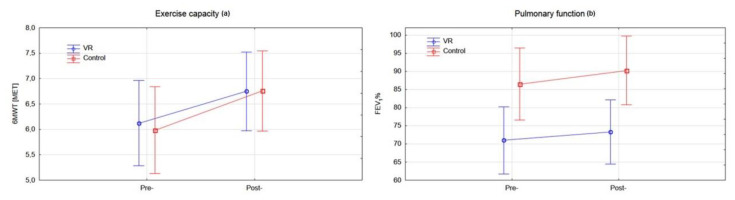
Analysis of the functional capacity in both groups pre- and post- rehabilitation program: (**a**) exercise capacity (**b**) pulmonary function.

**Table 1 jcm-10-00352-t001:** Baseline sociodemographic and clinical characteristics.

Variable	VR Group	Control Group	*p*
Age (years), mean (SD)	64.4 (5.7)	67.6 (9.4)	0.156
Female, *n* (%)	21 (84%)	20 (80%)	0.719
Body mass (kg), mean (SD)	77.3 (13.5)	73.8 (13.5)	0.355
Body high (cm), mean (SD)	162.3 (7.1)	164.1 (8.7)	0.436
BMI, mean (SD)	29.7 (5.3)	27.4 (3.8)	0.078
Time of tobacco consumption (years), mean (SD)	13.0 (17.0)	10.8 (13.0)	0.619
FEV1 pred., mean (SD)	71.0 (23.7)	86.5 (21.1)	0.047
6MWT (MET), mean (SD)	6.12 (1.97)	5.98 (1.84)	0.617
**Highest educational level, *n* (%)**			*p**
High school	5 (20%)	4 (16%)	0.677
College	14 (56%)	14 (56%)	
University	6 (24%)	7 (28%)	
**Material status, *n*(%)**			
Married/cohabiting	12 (48%)	14 (56%)	0.794
Divorced/widowed	13 (52%)	9 (36%)	
Single	0	2 (8%)	
**Employment status, *n* (%)**			
Professionally active	5 (20%)	5 (20%)	0.803
Retired	20 (80%)	20 (80%)	
**Subjective health status judgement, *n* (%)**			
Good	3 (12%)	7 (28%)	0.344
Neutral	18 (72%)	14 (56%)	
Bad	4 (16%)	4 (16%)	
**Voluntary physical activity, *n* (%)**			
Often	13 (52%)	15 (60%)	0.507
Occasionally	11 (44%)	10 (40%)	
None	1 (4%)	0	

6MWT: 6 min walk test; BMI: body mass index; FEV1 pred.: forced expiratory volume for 1 s predicted; MET: metabolic equivalent; SD: standard deviation; VR: virtual reality; *p*-Mann-Whitney U test; *p**-R rank Spearman.

**Table 2 jcm-10-00352-t002:** Analysis of the stress, anxiety, depression, and functional capacity assessment within the examined groups expressed as mean (SD).

Variable	VR Group			Control Group			Effect Size
	Pre	Post	*p*	Pre	Post	*p*	
Emotional tension	24.68 (6.12)	21.24 (6.03)	**0.0003**	22.44 (5.77)	21.28 (5.99)	0.1998	−0.377
External stress	19.24 (6.65)	17.12 (5.21)	**0.0092**	16.40 (4.71)	16.76 (4.77)	0.6470	−0.424
Intrapsychic stress	20.48 (5.85)	19.96 (5.78)	0.6356	17.52 (6.47)	17.76 (6.39)	0.8267	−0.121
PSQ Total score	64.40 (15.88)	58.32 (15.29)	**0.0069**	56.36 (14.91)	55.80 (14.19)	0.7961	−0.353
HADS-D mean (SD)	7.96 (2.76)	6.04 (3.21)	**0.0001**	6.64 (2.80)	7.08 (3.56)	0.4515	−0.836
HADS-A mean (SD)	10.36 (3.63)	8.24 (3.50)	**0.0009**	8.52 (3.22)	8.60 (2.87)	0.8941	0.631
HADS Total score	18.32 (4.90)	13.24 (4.05)	**0.0001**	15.16 (4.47)	15.68 (5.29)	0.6119	−1.175
6MWT [MET], mean (SD)	6.12 (2.12)	6.75 (2.24)	**0.0018**	5.98 (1.84)	6.76 (1.28)	**0.0002**	−0.074
FEV1 [%], mean (SD)	71.00 (23.66)	73.25 (23.24)	0.1893	86.48 (21.13)	90.24 (19.36)	**0.0429**	−0.066

Bold highlights statistical significance *p* < 0.05. 6MWT: 6 min walk test; FEV1: forced expiratory volume for 1 s; HADS: Hospital Anxiety and Depression Scale; MET: metabolic equivalent; SD: standard deviation; VR: virtual reality; PSQ: Perception of Stress Questionnaire.

**Table 3 jcm-10-00352-t003:** Predictors of stress levels (PSQ total score).

Determinant	Multiple Regression Model for Exercise Time R^2^ = 0.12, *p* < 0.00289	
	Standardized Beta Coefficient	*p*-Value
6MWT	−1.62	<0.043
FEV1	−0.17	<0.010

6MWT: 6-min walk test; FEV1: forced expiratory volume for 1 s.

**Table 4 jcm-10-00352-t004:** Predictors of depressive and anxiety symptoms (HADS Total score).

Determinant	Multiple Regression Model for Exercise Time R^2^ = 0.06, *p* < 0.019	
	Standardized Beta Coefficient	*p*-Value
FEV1	−2.44	<0.0189

FEV1: forced expiratory volume for 1 s.

## Data Availability

The data presented in this study are available on request from the corresponding author.

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
