# Peer review of "Evaluation of the Efficacy of Immersive Virtual Reality Therapy as a Method Supporting Pulmonary Rehabilitation: A Randomized Controlled Trial"

_jcm, 2021, doi:10.3390/jcm10020352_

Round 1

Reviewer 1 Report

I congratulate the authors for the results acquired in their study.
The subject is well described and the exposition is clear even to a reader unfamiliar with the topic.
I think that this paper could be very useful to improve the rehabilitation protocol specially in the era of COVID-19.
I suggest the authors to add some picture to better clarify the methodology of VR to the reader, such as an example of the first and last image showed to the patient (i.e. the grey garden and the more colourful one).

Author Response

Dear Reviewer, thank you for your comment.

I congratulate the authors for the results acquired in their study. The subject is well described and the exposition is clear even to a reader unfamiliar with the topic. I think that this paper could be very useful to improve the rehabilitation protocol specially in the era of COVID-19.

I suggest the authors to add some picture to better clarify the methodology of VR to the reader, such as an example of the first and last image showed to the patient (i.e. the grey garden and the more colourful one).

Thank you for that suggestion, we have supplemented the manuscript with the indicated images.

Reviewer 2 Report

This manuscript is a randomized clinical trial with 50 subjects who suffered from COPD in order to investigate the influence of adding the use of virtual reality to a traditional pulmonary rehabilitation program on anxiety and stress outcomes, aspects that usually do not receive the necessary attention in the management of these patients.

The strong points of the article are the detailed explanation of the methodology, as well as the innovation that the subject brings to this field of work.

Although I believe your work brings interesting data to the area of study, I am concerned about fewer aspects that the authors should consider:

Abstract: It is well written and adapted to the Consort checklist. The parts to be included are not detailed, but they are all included.

Introduction: It provides enough bibliographical references of the work needs on anxiety and stress in these patients, as well as on the evidence of virtual reality techniques.

However, other interventions to improve stress and anxiety in COPD patients have been conducted previously (Li Z, Liu S, Wang L, Smith L. Mind-Body Exercise for Anxiety and Depression in COPD Patients: A Systematic Review and Meta-Analysis. Int J Environ Res Public Health. 2019 Dec 18;17(1):22. doi: 10.3390/ijerph17010022), which should be included in the introduction and discussion, in order to discuss the greater or lesser effectiveness of these against the application of virtual reality.

It would be also interesting to include more explanation about effects that this combined therapy (rehabilitation + VR) has had on other population groups, in case there are such results in the literature.

Methods:

It is necessary that the authors make the statistical analysis following a General Linear Model that allows to analyze the differences in each variable in the complete model. Independent analysis of each group is not possible, since it would accumulate error.

Results:

Table 1. Since the subjects in the control group, without VR, start from a situation of FEV1 much higher (15% and with FEV1 values within normality) than that of the intervention group, it would be interesting to know the remaining values of the spirometric assessment, in order to compare their initial situation, since FEV1 alone is not indicative of suffering from COPDand the results obtained may be due to the different baseline situation of both groups and the different response to treatment, accordingly.

The authors must provide concrete data on the statistical significance of the general linear model, apart from the individual analyses they may make later.

Discussion:

In case the authors are not able to provide the information about the rest of the spirometric data indicated before, this should be indicated as a bias, since it may condition the response.

In addition, given the methodological bias in the statistical analysis, the conclusions obtained by the authors must be adjusted after carrying out the indicated statistical analysis.

I also consider that the authors should be more cautious when extrapolating the results to patients with COVID19 , and at most indicate that the effects of such type of intervention should be analyzed on COVID patients.

References: The bibliography is very current and agrees with the topic of study.

Minor revisions:

Line 21: p value has a mistake

CONSORT flow diagram: review assignment groups, the information is duplicated.

This is an article that may open up new ways of handling this type of patient, but some methodological errors make it necessary to review results and discussion.

Author Response

We thank the reviewer for the constructive comments on our manuscript. We have considered each of your comments and suggestions and we have made appropriate changes. Below are our responses to your remarks on a point-by-point basis.

1. Introduction: It provides enough bibliographical references of the work needs on anxiety and stress in these patients, as well as on the evidence of virtual reality techniques.
However, other interventions to improve stress and anxiety in COPD patients have been conducted previously (Li Z, Liu S, Wang L, Smith L. Mind-Body Exercise for Anxiety and Depression in COPD Patients: A Systematic Review and Meta-Analysis. Int J Environ Res Public Health. 2019 Dec 18;17(1):22. doi: 10.3390/ijerph17010022), which should be included in the introduction and discussion, in order to discuss the greater or lesser effectiveness of these against the application of virtual reality.

Thank you for pointing this out, we agree that a reference to the latest literature reviews is essential. However, in presented manuscript we quoted the indicated study in Line 41-44  of the introduction section and line 286-290 within the discussion section.

2. It would be also interesting to include more explanation about effects that this combined therapy (rehabilitation + VR) has had on other population groups, in case there are such results in the literature.

Thank you for this comment. The VR TierOne medical device was manufactured in 2019 with a grant from the Polish National Center of Research and Development (POIR.01-02.00-00-0134/16). Since then, clinical studies have been conducted to evaluate its effectiveness in: post-stroke rehabilitation (recorded at ClinicalTrials.gov under no. NCT03830372), cardiac rehabilitation (NCT04045977) and treatment of late-life depression (NCT04047511). In all cases, a decrease in stress levels, a reduction in anxiety and an improvement in well-being have been observed. The results of these studies are in the process of being published, but have not received a DOI number yet, and therefore cannot be quoted.

3. Methods:

It is necessary that the authors make the statistical analysis following a General Linear Model that allows to analyze the differences in each variable in the complete model. Independent analysis of each group is not possible, since it would accumulate error.

Thank you for pointing this out. We have performed two multivariable regression analyses to identify determinants of the depressive and anxiety symptoms and stress levels. Accordingly, we supplemented the description of statistical methods, results in the manuscript, and supplemented the discussions with the obtained results.

4. Results:

Table 1. Since the subjects in the control group, without VR, start from a situation of FEV1 much higher (15% and with FEV1 values within normality) than that of the intervention group, it would be interesting to know the remaining values of the spirometric assessment, in order to compare their initial situation, since FEV1 alone is not indicative of suffering from COPD and the results obtained may be due to the different baseline situation of both groups and the different response to treatment, accordingly.

Thank you for pointing this out. Unfortunately, we do not have comprehensive data for a full spirometry test. The qualification test prior the rehabilitation programme, are conducted in the Department of Functional Research of the MSWiA Hospital in Głuchołazy by a qualified laboratory employee. The criterion for inclusion in rehabilitation (COPD diagnosis) is the identification of the lower limit of the norm for the Tiffeneau-Pinelli index . Thus, the FEV1/FVC ratio for each patient was below 70% depending on body height and weight. Possibly, writing the index as FEV1% in the results table suggested Tiffeneau-Pinelli index, but it was FEV1 value expressed in %. We have improved the table entry to avoid such associations.

5. The authors must provide concrete data on the statistical significance of the general linear model, apart from the individual analyses they may make later.

Done, lines 223-232.

6. Discussion:

In case the authors are not able to provide the information about the rest of the spirometric data indicated before, this should be indicated as a bias, since it may condition the response.

In addition, given the methodological bias in the statistical analysis, the conclusions obtained by the authors must be adjusted after carrying out the indicated statistical analysis.

Thank you for pointing this out. The newest findings of the statistical analysis do not disturb the previously made conclusions. We have supplemented the study limitations with information about not including a full spirometry data analysis.

7. I also consider that the authors should be more cautious when extrapolating the results to patients with COVID19 , and at most indicate that the effects of such type of intervention should be analyzed on COVID patients.

Thank you for this comment. We are aware that the complications of COVID-19 concern, to a large extent, the functional state of the respiratory system, as well as the patient's psyche . Due to the very high level of stress associated with the COVID-19 pandemic, the strict isolation of patients during treatment and the threat of loss of life, many patients require the help of a psychologist or psychotherapist during recovery [29-32]. The effectiveness of virtual therapy in improving the mental health of patients with COPD, confirmed in this study, gives us reason to believe that this device can be successfully used in the rehabilitation of COVID-19 convalescents. We have already started studies to evaluate the effectiveness of VR TierOne in this group of patients. The first results support this argument, so we believe the statements regarding COVID-19 contained in the article to be appropriate. In addition, they have been included in the Discussion section, where authors have the right to discuss the potential significance of the obtained results, rather than in Conclusions which should relate directly to the research carried out.

8. Line 21: p value has a mistake

We apologize for this editorial error

9. CONSORT flow diagram: review assignment groups, the information is duplicated.

Thank you for pointing this out. We have adjusted the figure.

Round 2

Reviewer 2 Report

I would like to thank the authors for their efforts in modifying the manuscript and for taking into account most of the comments made in the review.

The application of RV in different population groups can mean an advance in different areas, among them the management of anxiety and stress, with results like those indicated by the authors.

After reviewing the authors' comments, and taking into account that the fact that they estimate the usefulness of the program for patients with COVID19 is something that is not obtained as a result in the study, it only remains that they adjust the conclusions in the abstract and eliminate the keyword COVID-19

I encourage the authors to continue in this line of work.

Author Response

Thank you very much for your suggestions that we believe significantly improved our manuscript.

We removed the keyword COVID-19, and changed the conclusions in the summary, lines 23-25.